# Pak1 kinase controls cell shape through ribonucleoprotein granules

Joseph O Magliozzi, James B Moseley*

Department of Biochemistry and Cell Biology, The Geisel School of Medicine at Dartmouth, Hanover, United States

**ABSTRACT** Fission yeast cells maintain a rod shape due to conserved signaling pathways that organize the cytoskeleton for polarized growth. We discovered a mechanism linking the conserved protein kinase Pak1 with cell shape through the RNA-binding protein Sts5. Pak1 (also called Shk1 and Orb2) prevents Sts5 association with P bodies by directly phosphorylating its intrinsically disordered region (IDR). Pak1 and the cell polarity kinase Orb6 both phosphorylate the Sts5 IDR but at distinct residues. Mutations preventing phosphorylation in the Sts5 IDR cause increased P body formation and defects in cell shape and polarity. Unexpectedly, when cells encounter glucose starvation, PKA signaling triggers Pak1 recruitment to stress granules with Sts5. Through retargeting experiments, we reveal that Pak1 localizes to stress granules to promote rapid dissolution of Sts5 upon glucose addition. Our work reveals a new role for Pak1 in regulating cell shape through ribonucleoprotein granules during normal and stressed growth conditions.

## Introduction

Cell polarity signaling networks determine cell morphology by controlling growth machinery in time and space. Because active growth requires energy, these networks must also respond to changes in the environment such as glucose availability. The p21-activated kinase (PAK) family of protein kinases are key mediators of cell polarity signaling in many eukaryotes (*Hofmann et al., 2004*). Following activation by small GTPases like Cdc42, PAKs function in regulating the cytoskeleton and MAPK pathways, and PAK dysregulation is associated with multiple human diseases. For example, PAKs regulate the elaborate architecture of highly polarized neurons, and PAK loss-of-function mutations impair neuronal migration, neurite outgrowth, and axonal development, leading to connections with autism spectrum disorders, Alzheimer's disease, neurofibromatosis type 1, and other neurological disorders (*Zhang et al., 2020*). Aberrant activation of human PAK1 also drives cell migration and proliferation in multiple forms of cancer including pancreatic, breast, prostate, and thyroid (*Bautista et al., 2020*; *Kanumuri et al., 2020*; *Rane and Minden, 2019*). Additionally, PAK1 plays a protective role in cardiac diseases including hypertrophy (*Wang et al., 2018*). The mechanisms that connect PAKs with these diseases include canonical functions mediated through Cdc42 and the cytoskeleton, along with MAPK pathways. However, the diversity of these disease-related defects also suggests additional noncanonical functions and mechanisms that remain less studied. The functions of PAKs in cell morphology and signaling are conserved in a wide range of cell types and organisms including yeast and humans (*Hofmann et al., 2004*). This conservation is demonstrated by the finding that expression of human PAK1 suppresses defects associated with loss of the PAK Ste20 in budding yeast (*Brown et al., 1996*). Full understanding of PAK function in cell polarity and proliferation requires identification of noncanonical substrates and functions for PAK-family proteins.

The fission yeast *Schizosaccharomyces pombe* is a strong model system to dissect conserved mechanisms controlling cell polarity and morphology due to its simple cylindrical shape, which can be modulated by environmental stresses and growth conditions. Early genetic screens identified different classes of fission yeast morphology mutants, such as round ('*orb*'), curved, bent, and branched (*Snell*

*For correspondence:
james.b.moseley@dartmouth.edu

**Competing interest:** The authors declare that no competing interests exist.

*and Nurse, 1994*; *Verde et al., 1995*). Genes that result in similar shape defects when mutated may function together in shared and/or overlapping pathways. In this study, we connect three factors identified as *orb* mutants – the protein kinases Pak1 and Orb6, along with the RNA-binding protein Sts5 – in a shared mechanism that controls fission yeast cell shape.

Fission yeast Pak1, which is also called Shk1 and Orb2, is an ortholog of human PAK1 and was initially identified as an *orb* mutant. Pak1 is essential for cell viability, and conditional *pak1* mutants become round and exhibit reduced viability during stress (*Marcus et al., 1995*; *Ottilie et al., 1995*; *Verde et al., 1998*). Many studies have linked Pak1 function with Cdc42 activation consistent with canonical PAK mechanisms (*Marcus et al., 1995*; *Ottilie et al., 1995*; *Tu and Wigler, 1999*). However, other studies identified Pak1 activation mechanisms that are independent of Cdc42 (*Yang et al., 1999*), consistent with similar activation mechanisms observed in other PAK kinases (*Lu et al., 1997*; *Shin et al., 2013*). Active Pak1 regulates components of the Cdc42 network and the cytoskeleton (*Edwards et al., 1999*; *Moshfegh et al., 2014*; *Vadlamudi et al., 2005*; *Vadlamudi et al., 2002*). Beyond these canonical targets, a recent phosphoproteomic screen identified potential substrates that suggest novel mechanisms for Pak1 function in cell morphology and viability (*Magliozzi et al., 2020*).

Similar to Pak1, the protein kinase Orb6 is a conserved regulator of cell morphology and was identified in genetic screens as an *orb* mutant (*Verde et al., 1995*). Orb6 is an essential NDR/LATS kinase that regulates cell polarity and morphology (*Verde et al., 1998*). Orb6 substrates include Cdc42 regulators and components of the exocyst, consistent with regulation of polarized growth (*Das et al., 2015*; *Das et al., 2009*; *Tay et al., 2019*). A recent study showed that Orb6 also phosphorylates Sts5, an RNA-binding protein that associates with processing bodies (P bodies) (*Nuñez et al., 2016*), which are ribonucleoprotein (RNP) granules that regulate translation and degradation of associated mRNAs. Conditional mutants in *orb6* and *pak1* are synthetically lethal (*Verde et al., 1998*; *Verde et al., 1995*), suggesting overlapping or connected pathways between these two protein kinases. However, specific connections in their signaling mechanisms have not been identified. In this study, we demonstrate that Sts5 phosphorylation by both Pak1 and Orb6 provides an additive regulatory mechanism for P body assembly, thereby connecting these two kinases through a shared substrate.

P bodies are biomolecular condensates comprising multiple proteins and mRNAs (*Eulalio et al., 2007*; *Luo et al., 2018*; *Parker and Sheth, 2007*; *Rao and Parker, 2017*). These cytoplasmic RNP structures are conserved from yeast through mammalian cells and regulate the translation and turnover of associated mRNAs. Fission yeast Sts5, which has similarity to human DIS3L2 and budding yeast Ssd1, associates with P bodies through an intrinsically disordered region (IDR) (*Jansen et al., 2009*; *Kurischko et al., 2011*). This association is partly inhibited by phosphorylation of the Sts5 IDR by Orb6 (*Chen et al., 2019*; *Nuñez et al., 2016*). Interestingly, Sts5 regulates mRNAs connected to cell polarity and was isolated in the same *orb* screens that identified Orb6 and Pak1, suggesting a shared cellular function (*Snell and Nurse, 1994*; *Verde et al., 1995*). Sts5 strongly localizes to RNP structures called stress granules when cells encounter glucose deprivation, thereby facilitating changes in both metabolism and growth (*Chen et al., 2019*; *Nuñez et al., 2016*). Stress granules assemble upon stress to regulate mRNA translation and stability. In yeast, stress granules and P bodies colocalize and may share some components (*Buchan and Parker, 2009*; *Protter and Parker, 2016*). The signaling mechanisms that connect Sts5 association to stress granules with glucose availability are largely unknown but have the potential to link cellular energy state with cell morphology.

In this work, we show that Pak1 directly phosphorylates Sts5 to regulate P bodies and cell morphology. Pak1 and Orb6 act synergistically on Sts5 to control cell shape through P body formation. Unexpectedly, glucose starvation induces rapid, PKA-dependent relocalization of Pak1 from the cell periphery to stress granules. Pak1 localization at stress granules is required for rapid dissolution of Sts5 from these structures when glucose is restored. Our results identify a novel nutrient-modulated mechanism connecting a conserved PAK kinase with cell morphology and stress response.

## Results

### RNA-binding protein Sts5 is a novel Pak1 substrate

We recently performed an unbiased phosphoproteomic screen that identified Sts5 as a leading candidate for phosphorylation by Pak1 in fission yeast cells (*Magliozzi et al., 2020*). Specifically,

phosphorylation of Sts5 residues S261 and S264 was reduced 2.5-fold in *pak1-as*, a partial loss-of-function mutant, compared to wildtype cells (*Loo and Balasubramanian, 2008*; *Magliozzi et al., 2020*). This phosphorylation was further reduced 2.3-fold by 15 min addition of 3-Brb-PP1, which completely inhibits *pak1-as* kinase activity (*Magliozzi et al., 2020*). S261 and S264 reside in the Sts5 IDR, which mediates its localization to RNP granules (*Chen et al., 2019*; *Nuñez et al., 2016*; *Toda et al., 1996*; *Vaggi et al., 2012*). To test if Pak1 directly phosphorylates Sts5, we conducted in vitro kinase assays using purified proteins and γ-$^{32}$P-ATP. Full-length Sts5 was directly phosphorylated by Pak1 but not by kinase-dead Pak1 (K415R) in vitro, confirming that Sts5 is a substrate of Pak1 (*Figure 1A and B*, *Figure 1—figure supplement 1*).

To test how phosphorylation by Pak1 might regulate Sts5 in cells, we imaged Sts5-mNeonGreen (mNG) in wildtype or *pak1-as* cells. In wildtype cells, Sts5 localized in a diffuse manner in the cytoplasm and formed very few granules (*Figure 1C and D*). In contrast, Sts5 localized to punctate granules in *pak1-as* cells (*Figure 1C and D*). These data show that Pak1 regulates the localization of its substrate Sts5. Consistent with a functional connection between Pak1 and Sts5, we observed a genetic interaction in the regulation of cell septation and separation. We measured an increased percentage of septated cells for *pak1-as* cells treated with 3-Brb-PP1, indicating a delay in cell separation, but this defect was suppressed by *sts5Δ* (*Figure 1E*). We conclude that Pak1 is required to prevent the aberrant formation of Sts5 granules, which can interfere with cell separation and potentially other cellular processes.

Dysregulation of Sts5 in *pak1-as* could reflect direct kinase-substrate regulation or alternatively indirect consequences of altered Pak1 signaling. To distinguish between these possibilities, we generated a non-phosphorylatable Sts5(S261A S264A) mutant based on the two sites modified by Pak1 in cells. The Sts5(S261A S264A) mutant lost phosphorylation by Pak1 in vitro and constitutively localized to puncta in cells (*Figure 1F, G and H*). This localization defect was phenocopied by the single S261A mutant but not by S264A (*Figure 1G and H*), demonstrating that S261 is the primary site for regulation by Pak1. These combined data indicate that Pak1 phosphorylates the Sts5 IDR to prevent its localization to punctate granules in the cytoplasm.

## Pak1-Orb6-Sts5 regulatory axis controls polarized growth and cell separation

Recent studies have shown that NDR/LATS kinase Orb6 similarly phosphorylates the Sts5 IDR to regulate its localization to RNP granules (*Figure 2A*; *Chen et al., 2019*; *Nuñez et al., 2016*). Importantly, Orb6 phosphorylates S86 in the Sts5 IDR (*Chen et al., 2019*), while we have shown that Pak1 directly phosphorylates S261. These distinct sites of phosphorylation raise the possibility that Pak1 and Orb6 perform non-redundant functions on Sts5. To test this possibility, we made non-phosphorylatable mutations either at the Pak1 and Orb6 sites alone (S86A or S261A), or at both sites in combination (S86A S261A, which we refer to as Sts5-2A). Using SDS-PAGE containing Phosbind, which slows the migration of phosphorylated proteins, we found dramatic loss of phosphorylation for Sts5-2A compared to wildtype or either single mutant alone (*Figure 2B*, *Figure 2—figure supplement 1*). Thus, mutating the primary phosphorylation sites for Pak1 and Orb6 in combination causes loss of Sts5 phosphorylation in cells.

Preventing Sts5 phosphorylation by Pak1 and Orb6 had dramatic effects on Sts5 localization and function. While either Sts5-S86A or Sts5-S261A alone increased the number of Sts5 granules per cell, combining these mutations in Sts5-2A substantially increased granule number beyond either single mutant alone (*Figure 2C and D*). Increased Sts5 granule localization correlated with defects in cell morphology, septation, and viability. The single mutants S86A and S261A exhibited increased cell width compared to wildtype cells, but the double mutant Sts5-2A significantly increased cell width over either single mutant alone (*Figure 2E and F*). This increased width is consistent with a partial *orb* phenotype for the Sts5-2A mutant, which exhibited additional cell polarity defects including an increased fraction of monopolar cells (*Figure 2—figure supplement 2A, B*). Beyond cell polarity, the septation index of Sts5-2A was significantly increased over wildtype and the single mutants, consistent with a defect in cell separation (*Figure 2G*). Finally, the viability of Sts5-2A mutants was dramatically decreased under a variety of stress conditions compared to wildtype and the single mutants (*Figure 2H*, *Figure 2—figure supplement 3*). In conclusion, Pak1 and Orb6 kinases function additively on the Sts5 IDR to control whether Sts5 localizes to the cytoplasm or to granules. Defects in this



**Figure 1.** Pak1 directly phosphorylates RNA-binding protein Sts5 to control its localization. (**A**) Schematic of Sts5 domain layout. Pak1-dependent phosphorylation sites are indicated by red lines. IDR: intrinsically disordered region; CSD: cold shock domain. (**B**) In vitro kinase assay with purified proteins and γ-$^{32}$P-ATP. (**C**) Sts5-mNG localization in wildtype versus *pak1-as* cells. (**D**) Quantification of the number of Sts5-mNG granules per cell. Values are mean ± SD (n = 50 cells per strain). ****p<0.0001. (**E**) Quantification of septation index of indicated strains. Values are mean ± SD from three

*Figure 1 continued on next page*

*Figure 1 continued*

biological replicates (n > 150 cells each). ***p<0.001; **p<0.01. (**F**) In vitro kinase assay with purified proteins and γ-$^{32}$P-ATP. GST-Sts5 protein loading was assessed by SDS-PAGE followed by silver staining. (**G**) Sts5-mNG localization in indicated strains. (**H**) Quantification of Sts5-mNG granule number per cell. Values are mean ± SD (n = 50 cells per strain). ****p<0.0001. n.s.: not significant. Images in (**C**) and (**G**) are maximum intensity projections from spinning disc confocal microscopy. Scale bars, 5 µm.

The online version of this article includes the following figure supplement(s) for figure 1:

**Source data 1.** Raw values shown on graphs in *Figure 1D,E and H*.

**Source data 2.** Original files of the full raw unedited gels in *Figure 1*.

**Source data 3.** Original files of the full raw unedited gels in *Figure 1*.

**Source data 4.** Original files of the full raw unedited gels in *Figure 1*.

**Figure supplement 1.** Purified proteins for in vitro kinase assays.

**Figure supplement 2.** Uncropped versions of gel images in *Figure 1*.

combined regulatory mechanism lead to aberrant Sts5 granules, along with additive defects in cell morphology, division, and viability.

## Downstream effects of Sts5 phosphorylation on P bodies and protein levels

Sts5 is homologous to human Dis3L2, a 3'–5' exoribonuclease, but Sts5 lacks the essential catalytic residues to degrade mRNA targets. Past studies have shown that Sts5 binds to mRNAs and targets them to P bodies, which contain machinery to reduce mRNA levels and translation in cells (*Malecki et al., 2013*; *Nuñez et al., 2016*; *Uesono et al., 1997*). Several transcripts related to cell morphology, including the protein kinases Ssp1 and Cmk2, are major targets of Sts5 regulation (*Nuñez et al., 2016*). Sts5 localizes to P bodies through its IDR, consistent with the role of such domains in association with biomolecular condensates through a process akin to liquid-liquid phase separation (*Alberti et al., 2019*). In many cases, phosphorylation within an IDR prevents accumulation in condensates (*Owen and Shewmaker, 2019*). We hypothesized that phosphorylation by Pak1 and Orb6 could prevent Sts5 clustering and association with P bodies, thereby maintaining full expression of its targets such as Ssp1 and Cmk2. To test this idea, we performed colocalization experiments with Sts5-mNG and Sum2-tdTomato. Sum2 is the *S. pombe* ortholog of budding yeast Scd6, a protein translation inhibitor that is a core component of P bodies (*Nissan and Parker, 2008*; *Rajyaguru et al., 2012*). Wildtype Sts5-mNG rarely formed granules and therefore did not colocalize with Sum2 in P bodies (*Figure 3A and B*). In contrast, Sts5-2A-mNG formed granules that colocalized with Sum2 and increased the number of P bodies per cell (*Figure 3A and B*). We conclude that additive phosphorylation of the Sts5 IDR by Pak1 and Orb6 prevents its clustering and association with P bodies.

A previous study showed that Orb6 phosphorylation of Sts5-S86 promoted binding of the 14-3-3 protein Rad24, which prevents Sts5 clustering (*Nuñez et al., 2016*). Sts5-S261 also fits the minimal consensus motif for binding to 14-3-3 (R-X-X-pS). Using co-immunoprecipitation (*Figure 3C and D* and *Figure 3—figure supplement 1*), we found that the S261A mutation decreased Sts5 interaction with Rad24 similarly to the S86A. Importantly, the double mutant Sts5-2A nearly abolished binding to Rad24, showing that Pak1 and Orb6 act in an additive manner to prevent Sts5 clustering through phosphorylation-dependent binding to Rad24.

Our results also indicate that clustering of Sts5 induces formation of P bodies, which we predicted would decrease the levels of Sts5 targets including Ssp1 and Cmk2. Consistent with this prediction, the levels of Ssp1 and Cmk2 protein were significantly reduced in Sts5-2A cells as compared to either single mutant alone or to wildtype cells (*Figure 4A, B, D and E*). Additionally, the level of mRNA transcripts for Ssp1 and Cmk2 was significantly reduced in Sts5-2A cells compared to wildtype cells (*Figure 4C and F*). Other known Sts5 mRNA-binding targets Psu1 and Efc25 transcript levels were also significantly reduced in Sts5-2A cells compared to wildtype cells (*Figure 4—figure supplement 1A, B*). Put together, these results demonstrate that loss of Sts5 phosphorylation by Pak1 and Orb6 prevents Sts5-Rad24 binding, leading to aberrant P body assembly and reduced levels of Ssp1 and Cmk2 (*Figure 4G*). This mechanism connects with changes in cell morphology and survival under stress.

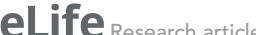

**Figure 2.** Pak1 and Orb6 kinases phosphorylate distinct residues in the Sts5 intrinsically disordered region (IDR) for cell morphology. (**A**) Schematic of Sts5 domain layout. Pak1 and Orb6 phosphorylation sites are indicated by red lines. CSD: cold shock domain. (**B**) Western blot of whole-cell extracts from the indicated strains separated by SDS-PAGE containing Phosbind. Note loss of Sts5 phosphorylation in the 2A mutant, which contains both S86A and S261A mutations. Ponceau stain included for loading. (**C**) Spinning disc confocal images of Sts5-mNG localization. Images are maximum intensity

*Figure 2 continued on next page*

*Figure 2 continued*

projections. (**D**) Quantification of Sts5-mNG granule number per cell. Values are mean ± SD (n = 50 cells per strain). ****p<0.0001. n.s.: not significant. (**E**) Images of Sts5-mNG cells stained with Blankophor. (**F**) Quantification of cell width in Sts5-mNG strains. Values are mean ± SD (n ≥ 50 cells per strain). ****p<0.0001; ***p<0.001. (**G**) Quantification of septation index of indicated strains. Values are mean ± SD from three biological replicates (n > 150 cells each). ****p<0.0001. n.s.: not significant. (**H**) Serial-dilution growth assays for each indicated strain. Strains were spotted onto indicated media and grown for 3–5 days at either 32°C or 37°C. Scale bars, 5 µm.

The online version of this article includes the following figure supplement(s) for figure 2:

**Source data 1.** Raw values shown on graphs in *Figure 2D, F and G*.

**Source data 2.** Original files of the full raw unedited gels in *Figure 2*.

**Source data 3.** Original files of the full raw unedited gels in *Figure 2*.

**Figure supplement 1.** Phosbind western blot of all Sts5 phosphomutants.

**Figure supplement 2.** Sts5 phosphomutants display defects in cell polarity.

**Figure supplement 2—source data 1.** Raw values shown on graph in *Figure 2—figure supplement 2B*.

**Figure supplement 3.** Mutation of Sts5 residue S264 does not impact cell growth.

**Figure supplement 4.** Uncropped versions of gel images from *Figure 2*.

## Glucose starvation promotes Pak1 localization to stress granules

Pak1 and Orb6 phosphorylate Sts5 at distinct residues to regulate P body formation in an additive manner. Why would two different kinases be required for this regulatory mechanism? We considered the possibility that Pak1 and Orb6 might respond to distinct physiological states to link Sts5 regulation with a range of growth conditions. Orb6 regulation of Sts5 has been connected to both cell cycle progression and nitrogen starvation (*Chen et al., 2019*; *Vaggi et al., 2012*). We noted that glucose starvation serves as a signal for enhanced recruitment of Sts5 to stress granules in a manner that is reversed by reintroduction of glucose. Therefore, we tested the localization of Pak1, Orb6, and Sts5 during glucose starvation. Sts5 localized to large granules consistent with previous work (*Figure 5A*; *Nuñez et al., 2016*). Interestingly, Pak1 was similarly concentrated into cytoplasmic granules, but Orb6 remained diffuse in the cytoplasm (*Figure 5A*, *Figure 5—figure supplement 1A*). To determine the nature of these granules, we tested their colocalization with the P body marker Sum2 and with the stress granule marker Pabp. P bodies are constitutive structures, while stress granules assemble during under specific conditions such as glucose deprivation (*Corbet and Parker, 2019*; *Luo et al., 2018*; *Protter and Parker, 2016*). P bodies and stress granules overlap and may share components during stress conditions, particularly in yeast (*Buchan et al., 2008*; *Kedersha et al., 2005*; *Protter and Parker, 2016*). Both Pak1 and Sts5 granules colocalized with Sum2 and Pabp (*Figure 5B and C*, *Figure 5—figure supplement 2*), and Pak1 and Sts5 colocalized together at granules (*Figure 5D*). We refer to Pak1 and Sts5 localization to stress granules during glucose deprivation due to colocalization with Pabp, but we note that this result formally supports association with the overlapping stress granule and P body structures. These data show that glucose starvation induces a change in Pak1 localization away from growing cell tips and into stress granules, which contain Sts5 but not Orb6. Pak1 localization to stress granules was independent of Sts5 as we observed this localization pattern in both *sts5Δ* and *sts5-2A* mutant cells during glucose starvation (*Figure 5E*, *Figure 5—figure supplement 1B, C*).

We next tested if Pak1 recruitment to stress granules was specific to glucose starvation or alternatively represented a general stress response. Pak1 did not form cytoplasmic clusters in nitrogen starvation or hyperosmotic stress (*Figure 5F*). We observed Pak1 clusters during heat stress; these clusters did not colocalize with Sum2 (*Figure 5F*) but partially colocalized with Pabp (*Figure 5—figure supplement 1D*). Thus, Pak1 strongly associates with stress granules during glucose starvation but not during all environmental stresses. We tested if other cell polarity kinases, in addition to Pak1, also localized to stress granules during glucose starvation. For this experiment, we screened the localization of a panel of fluorescently tagged kinases in high (2%) versus low (0%) glucose conditions. Pck2 and Kin1 localized to cytoplasmic clusters during glucose starvation (*Figure 5—figure supplement 3A, B*). We note that Pck2 was recently shown to localize to stress granules during high heat stress (*Kanda et al., 2020*). In contrast, Orb6, Nak1, and Pck1 did not form clusters during glucose starvation (*Figure 5A*, *Figure 5—figure supplement 3A*). We conclude that glucose starvation induces the



**Figure 3.** Sts5 granules associate with P bodies in a 14-3-3 protein Rad24-dependent manner. (**A**) Localization of Sts5 and Sum2. (**B**) Quantification of Sts5 granule number per cell (top), P body number per cell (middle), and number of colocalized Sts5 granules and P bodies (bottom). Values are mean ± SD (n = 35 cells per strain). ****p<0.0001. (**C**) Representative co-immunoprecipitation of Rad24-GFP and indicated Sts5-FLAG strains. (**D**) Quantification of immunoprecipitated Sts5-FLAG protein levels from western blotting. Values are mean ± SD from three biological replicates. ***p<0.001; **p<0.01. n.s.: not significant. Images in (**A**) and (**B**) are spinning disc confocal maximum intensity projections of the top half of cells. Insets are enlarged views of boxed regions. Dotted lines show cell outlines. Scale bars, 5 µm.

The online version of this article includes the following figure supplement(s) for figure 3:

**Source data 1.** Raw values shown on graphs in *Figure 3B and D*.

**Source data 2.** Original files of the full raw unedited gels in *Figure 3*.

*Figure 3 continued on next page*

*Figure 3 continued*
**Source data 3.** Original files of the full raw unedited gels in *Figure 3*.
**Figure supplement 1.** Physical association of Rad24 and Sts5.
**Figure supplement 2.** Uncropped versions of gel images from *Figure 3*.

redistribution of Pak1 away from sites of polarized growth and into stress granules, where it colocalizes with its substrate Sts5 and the P body marker Sum2. Comparing Pak1 and Orb6, this association with stress granules is specific to Pak1 and identifies a clear distinction between these two regulators of Sts5.

For polarized growth in rich media conditions, Pak1 functions in a regulatory module that promotes Cdc42 activation at growing cell tips. This module includes the Pak1 substrate Scd1, which is a guanine nucleotide exchange factor (GEF) for Cdc42 (*Chang et al., 1994*), and the scaffold protein Scd2, which binds to both Pak1 and Scd1 (*Chang et al., 1999*; *Endo et al., 2003*). In contrast to Pak1, Scd2 and Scd1 did not concentrate at Sum2-marked granules during glucose starvation (*Figure 6A and B*). Similarly, we did not observe granule localization for Cdc42 containing a functional fluorescent tag or for the well-characterized active Cdc42 biomarker CRIB-3xGFP (*Figure 6C and D*). These results indicate that Pak1 localizes to stress granules during glucose starvation independently of its cell polarity regulatory module.

## PKA signaling is required for Pak1 stress granule localization

Localization of Pak1 to stress granules specifically during glucose starvation suggests the involvement of a glucose-modulated signaling pathway. We tested the role of proteins that signal changes in glucose availability and cellular energy status. During glucose starvation, Pak1 still colocalized with Sum2 upon deletion of Ssp2, the catalytic subunit of AMP-activated protein kinase, and upon deletion of its upstream activator Ssp1 (*Deng et al., 2017*; *Schutt and Moseley, 2017*; *Valbuena and Moreno, 2012*; *Figure 7A*). Similarly, Pak1 colocalized with Sum2 after deletion of Sds23 (*Figure 7A*), which regulates glucose-dependent phosphatase activity (*Hanyu et al., 2009*; *Ishii et al., 1996*). In contrast, Pak1 was absent from granules in *pka1Δ* cells (*Figure 7B*). Pka1 is the catalytic subunit of protein kinase A (PKA), which regulates cell cycle regulation and stress granule assembly during glucose deprivation in yeast (*Kelkar and Martin, 2015*; *Nilsson and Sunnerhagen, 2011*). If PKA signaling mediates Pak1 localization, then other mutations in the PKA pathway should phenocopy *pka1Δ*. Consistent with this prediction, Pak1 did not localize to stress granules during glucose starvation in *git3Δ*, *gpa2Δ*, or *cyr1Δ* mutants, which act in a linear activation pathway for PKA activation (*Hoffman, 2005*; *Kelkar and Martin, 2015*; *Figure 7B*). Importantly, both Sts5 and Sum2 still formed granules during glucose deprivation in all of these PKA pathway mutants, which means that Pak1 localization defects are not due to overall loss of RNP granules (*Figure 7C*). These data identify a PKA-mediated pathway that is required for localization of Pak1 to stress granules during glucose starvation.

## Pak1 functions to promote stress granule dissolution

Finally, we sought to define a functional role for Pak1 localization to stress granules during glucose starvation. Because Pak1 phosphorylates the Sts5 IDR to inhibit its clustering and association with P bodies, we hypothesized that Pak1 might promote Sts5 dissociation from stress granules upon glucose addition. To test this hypothesis, we aimed to prevent Pak1 localization to stress granules. PKA signaling has multiple effects on RNP granules (*Nilsson and Sunnerhagen, 2011*), so we needed a more direct method to disrupt Pak1 localization specifically. We found that addition of a lipid-modified CAAX motif forced constitutive cortical localization of Pak1, resulting in increased cell width (*Figure 8A*, *Figure 8—figure supplement 1A, B*). In high glucose conditions, Pak1-GFP-CAAX localized to the cell cortex and concentrated at cell tips (*Figure 8A*, *Figure 8—figure supplement 1A*). During glucose starvation, Pak1-GFP-CAAX failed to redistribute to stress granules and instead localized throughout the cell cortex (*Figure 8A*, *Figure 8—figure supplement 1A*).

We used this synthetic retargeting tool to test how Pak1 at RNP granules regulates Sts5 dissociation from these structures. We monitored the dissociation kinetics by measuring the concentration of Sts5-mNG per RNA granule following addition of glucose. In wildtype cells, Sts5 intensity rapidly dropped and plateaued to background levels by 5–6 min after addition of glucose (*Figure 8C*). In



**Figure 4.** Formation of Sts5 granules reduces mRNA and protein levels of Ssp1 and Cmk2. (**A**) Representative western blot showing Ssp1 protein levels in the indicated strains. α-Cdc2 blot shown as loading control. (**B**) Quantification of Ssp1 protein levels from western blotting. Values are mean ± SD from three biological replicates. **p<0.01. n.s.: not significant. (**C**) Normalized counts for Ssp1 transcripts from the indicated strains using NanoString gene expression assays. Values are mean ± SD from three biological replicates. ***p<0.001. (**D**) Representative western blot showing Cmk2 protein levels in the indicated strains. α-Cdc2 blot shown as loading control. (**E**) Quantification of Cmk2 protein levels from western blotting. Values are mean ± SD from three biological replicates. **p<0.01. n.s.: not significant. (**F**) Normalized counts for Cmk2 transcripts from the indicated strains using NanoString gene expression assays. Values are mean ± SD from three biological replicates. **p<0.01. (**G**) Model for Sts5 phosphorylation by Pak1 and Orb6 to control both P body formation and cell shape.

The online version of this article includes the following figure supplement(s) for figure 4:

**Source data 1.** Raw values shown on graphs in *Figure 4B, C, E and F*.

**Source data 2.** Original files of the full raw unedited gels in *Figure 4*.

**Source data 3.** Original files of the full raw unedited gels in *Figure 4*.

**Source data 4.** Original files of the full raw unedited gels in *Figure 4*.

**Source data 5.** Original files of the full raw unedited gels in *Figure 4*.

**Figure supplement 1.** Sts5 granules result in decreased Psu1 and Efc25 mRNA levels.

**Figure supplement 2.** Uncropped versions of gel images from *Figure 4*.

**Figure 5.** Pak1 localizes to stress granules with Sts5 during glucose starvation. (**A**) Localization of the indicated protein kinases in the presence or absence of glucose. Images are single medial focal planes. Arrows point to granule localization of either Sts5 or Pak1. (**B**) Localization of Pak1 and Sum2 during glucose starvation. (**C**) Localization of Sts5 and Sum2 during glucose starvation. (**D**) Localization of Pak1 and Sts5 during glucose starvation. (**E**) Localization of Pak1 and Sum2 in glucose-starved *sts5Δ* cells. Images in (**B**–**E**) are spinning disc confocal maximum intensity projections. (**F**) Localization of Pak1 and Sum2 under different stress conditions. Images are maximum intensity projections. Insets are enlarged views of boxed regions. Dotted lines show cell outlines. Scale bars, 5 μm.

The online version of this article includes the following figure supplement(s) for figure 5:

*Figure 5 continued on next page*

*Figure 5 continued*

**Figure supplement 1.** Localization of Pak1 and Orb6 in the presence and absence of glucose.

**Figure supplement 2.** Sts5, Pak1, and Sum2 colocalize with the stress granule marker Pabp during glucose starvation.

**Figure supplement 3.** Localization of other polarity kinases during glucose starvation.

contrast, the rate of Sts5 loss from granules was reduced in Pak1-CAAX cells. Importantly, the levels of Sts5 at granules were identical in Pak1-CAAX and control cells at the time of glucose addition (*Figure 8C*). However, Sts5 dissociated more slowly in Pak1-CAAX cells, and Sts5 levels did not reach background levels until 10 min in Pak1-CAAX (*Figure 8B and C*). Reduced Sts5 dissociation kinetics in Pak1-CAAX cells was observed in multiple experimental replicates (*Figure 8—figure supplement 1C, D*) and also caused a delay in reducing the number of Sts5 granules following addition of glucose (*Figure 8D*). We conclude that PKA-dependent Pak1 localization to stress granules promotes rapid dissociation of its substrate Sts5 upon glucose addition (*Figure 8E*).

## Discussion

The main findings from our study are (1) Pak1 directly phosphorylates the Sts5 IDR to prevent recruitment to P bodies, (2) Pak1 and Orb6 kinases act in an additive manner on Sts5 to control cell morphology and repression of Ssp1, (3) Pak1 is recruited to stress granules with Sts5 during glucose starvation through a PKA-dependent pathway, and (4) Pak1 localization to stress granules promotes

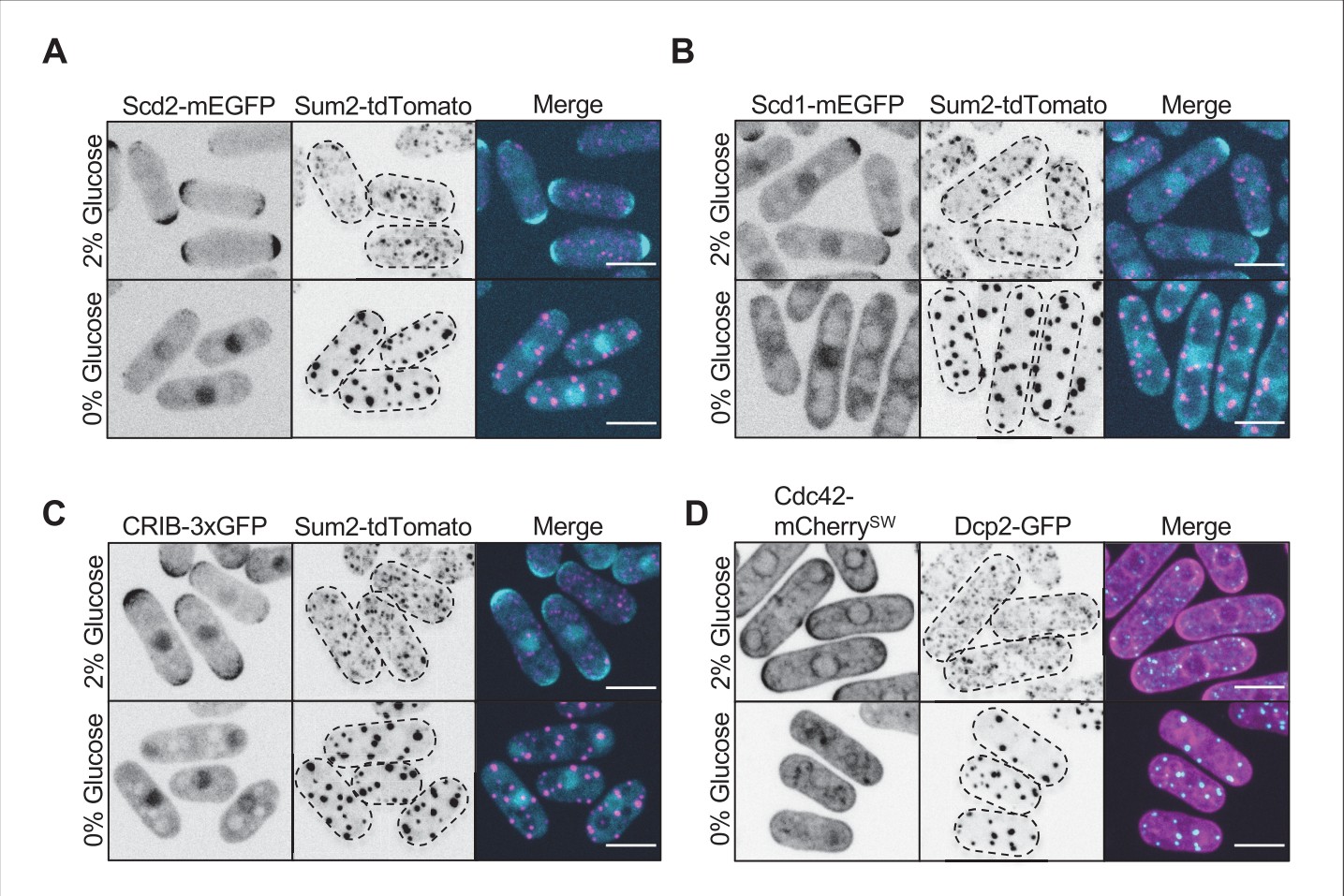

**Figure 6.** Pak1 ligands in Cdc42 regulatory pathway do not localize to stress granules. (**A**) Localization of Scd2 and Sum2. (**B**) Localization of Scd1 and Sum2. (**C**) Localization of the activated Cdc42 biosensor CRIB-3xGFP and Sum2. (**D**) Localization of Cdc42 and Dcp2. Images are spinning disc confocal maximum intensity projections. Dotted lines show cell outlines. Scale bars, 5 μm.



**Figure 7.** PKA signaling mediates Pak1 recruitment to stress granules during glucose starvation. (**A**) Localization of Pak1 and Sum2 for indicated strains in 0% glucose. (**B**) Localization of Pak1 and Sum2 for indicated strains in 0% glucose. Note the loss of Pak1 localization to stress granules in the mutant strains. (**C**) Localization of Sts5 and Sum2 for indicated strains in 0% glucose. Images are spinning disc confocal maximum intensity projections. Dotted lines show cell outlines. Scale bars, 5 µm.



**Figure 8.** Pak1 promotes rapid Sts5 dissolution from stress granules upon glucose addition. (**A**) Localization of Pak1-CAAX and Sum2. Images are spinning disc confocal maximum intensity projections. Dotted lines show cell outlines. (**B**) Localization of Sts5 in indicated strains. Images are spinning disc confocal sum projections. (**C**) Quantification of Sts5-mNG fluorescence intensity per stress granule following glucose refeed for indicated strains. Values are means ± SD (n = 50 granules per timepoint). ****p<0.0001. n.s.: not significant. Scale bars, 5 μm. (**D**) Quantification of number of Sts5-mNG granules per cell following glucose refeed for indicated strains. Values are means ± SD (n = 50 cells per timepoint). ****p<0.0001; ***p<0.001. n.s.: not

*Figure 8 continued*

significant. Scale bars, 5 µm. (**E**) Model showing PKA-dependent Pak1 localization to stress granules to promote rapid Sts5 dissolution during glucose refeed.

The online version of this article includes the following figure supplement(s) for figure 8:

**Source data 1.** Raw values shown on graphs in *Figure 8C and D*.

**Figure supplement 1.** Pak1 functions at stress granules to promote Sts5 dissolution.

**Figure supplement 1—source data 1.** Raw values shown on graphs in *Figure 8—figure supplement 1B, C and D*.

rapid dissolution of Sts5 from these structures upon addition of glucose. These results identify and define a new, bidirectional connection between PAK and RNP granules. This connection ensures proper cell morphology during nutrient-rich conditions and rewires PAK signaling during glucose starvation.

## The Orb polarity module

We identified Sts5 as a novel Pak1 substrate through an unbiased phosphoproteomic screen (*Magliozzi et al., 2020*) followed by targeted studies in vitro and in vivo. PAKs are classically known to regulate cell morphology and signaling through Cdc42 and MAPK pathways, but our results demonstrate that Pak1 also acts through Sts5 and P bodies, thereby expanding Pak1 function in cell polarity through a glucose-dependent mechanism. This connection between a PAK and RNP granules provides a new mechanism that may operate in other organisms as well.

We found that Pak1 acts together with the NDR kinase Orb6 on Sts5. Both kinases phosphorylate the Sts5 IDR but at distinct residues: S86 for Orb6 (*Chen et al., 2019*) and S261 for Pak1. Preventing these modifications with the Sts5-2A mutant led to constitutive P body formation, increased cell width, and decreased viability during stress. Our work supports a model where phosphorylation of the Sts5 IDR by both Pak1 and Orb6 prevents Sts5 incorporation into P bodies. Sts5 binds and recruits mRNAs including *ssp1* into P bodies (*Nuñez et al., 2016*). Loss of regulation by Pak1 and Orb6 leads to decreased levels of Ssp1 and Cmk2 protein, consistent with translational repression and/or mRNA turnover at P bodies. In this model, phosphorylation by Pak1 and Orb6 may prevent Sts5 aggregates through multiple mechanisms. Past work has shown that phosphorylation by Orb6 enhances Sts5 binding to Rad24, a 14-3-3 protein that sequesters Sts5 away from P bodies (*Chen et al., 2019*; *Nuñez et al., 2016*). Similarly, we found that mutating both the Orb6 and Pak1 phosphorylation sites dramatically reduced Sts5-Rad24 interactions. We note that phosphorylation of proteins with IDRs appears to inhibit their incorporation into biomolecular condensates through liquid-liquid phase separation both in vitro and in cells (*Owen and Shewmaker, 2019*), meaning that phosphorylation of Sts5 IDR may contribute to both Rad24-dependent and -independent forms of regulation. More generally, the connection between RNPs and cell polarity is an emerging theme that has been described in several organisms (*Gerbich et al., 2020*; *Jansen et al., 2009*; *Lee et al., 2015*; *Lee et al., 2020*). Our work supports a critical role for regulated assembly of P bodies in the cell polarity program that establishes and maintains cell morphology. Given the conservation of PAKs, NDR kinases, and P bodies, our work has implications for regulation of cell polarity by RNPs in other cell types and organisms.

In this mechanism, Pak1 and Orb6 appear to play very similar roles in regulation of Sts5, raising the question of why this system uses two distinct upstream kinases. We found that Pak1 but not Orb6 localizes to stress granules upon glucose starvation. This localization was not observed in other stresses or during specific cell cycle stages. Meanwhile, Orb6 activity is cell cycle-regulated, and Orb6 signaling to Sts5 becomes important during nitrogen stress (*Chen et al., 2019*; *Nuñez et al., 2016*). Put together, these observations indicate that Pak1 and Orb6 respond to distinct physiological cues in their regulation of Sts5. In some cases, both Pak1 and Orb6 are active and likely to regulate Sts5 in an additive manner. In other cases, either Pak1 or Orb6 alone is active. By using protein kinases that respond differently to various cues, this mechanism may tune P body regulation to a range of physiological conditions.

We note that Pak1 (also called Orb2), Sts5 (also called Orb4), and Orb6 were all identified in the same genetic screen for *orb*-shaped mutant (*Snell and Nurse, 1994*; *Verde et al., 1995*). Our results unite these three factors in a shared regulatory mechanism. We observed increased cell width in the *sts5-2A* mutant, which loses phosphorylation by Pak1 and Orb6. This phenotype represents a partial

*orb* shape, confirming its role in this process but also indicating that additional factors contribute to the complete *orb* shape defect. Consistent with this possibility, both Pak1 and Orb6 have additional substrates within the Cdc42 and exocytosis networks (*Chang et al., 1999*; *Das et al., 2015*; *Magliozzi et al., 2020*; *Tay et al., 2019*). Pak1 and Orb6 may control cell polarity through coordinated regulation of RNPs and Cdc42 signaling, or alternatively by acting independently on these two downstream targets. Additionally, the connection between Pak1, Orb6, and Sts5 raises the possibility that other *orb* genes also function in this mechanism. For example, Orb5 is also called Cka1, the catalytic subunit of casein kinase 2 (CK2) (*Snell and Nurse, 1994*). In human cells, CK2 phosphorylates and activates PAK1 (*Shin et al., 2013*), but this connection has not been tested in fission yeast to our knowledge. Future studies may reveal the involvement of other Orb proteins in this shared regulatory mechanism centered on phospho-regulation of RNPs.

## Regulation and function of Pak1 at stress granules

Our results show that Pak1 is recruited to stress granules during glucose starvation, raising two important questions. First, how is Pak1 recruited to stress granules? We found that Pak1 localization to stress granules is independent of Sts5 but requires PKA signaling. These combined data suggest that Pak1 binds to an unknown factor within RNPs, and this interaction depends on PKA. We note that PKA phosphorylates PAK1 in human cells (*Howe and Juliano, 2000*), raising the possibility of direct regulation. However, PKA has multiple effects on RNP granule assembly during stress and is required for stress granule assembly during glucose deprivation (*Nilsson and Sunnerhagen, 2011*). The effect of PKA on Pak1 localization may work through these broader effects on stress granules or alternatively on PKA-dependent phosphorylation of other stress granule proteins that bind to Pak1. Future work on PKA substrates within RNP granules could reveal how Pak1 localizes to these sites in a regulated manner.

A second important question is the function of Pak1 at stress granules. Pak1-CAAX prevented localization to granules and led to slower dissolution of Sts5 upon glucose addition. This result suggests that Pak1 at stress granules is poised to promote rapid release of its substrate Sts5 as cells re-enter polarized growth. This model is consistent with several other protein kinases that localize to RNP granules during specific stress conditions and function to dissolve granules upon removal of stress (*Rai et al., 2018*; *Shattuck et al., 2019*; *Wippich et al., 2013*; *Yahya et al., 2021*). Alternatively, stress granules could represent a storage depot for inactive Pak1 when cells stop growth during glucose starvation. In support of this model, PKA inhibits PAK1 kinase activity in human cells (*Howe and Juliano, 2000*), and several other protein kinases have been shown to be sequestered at RNPs during various stress conditions (*Kanda et al., 2020*; *Shah et al., 2014*). Future studies will be directed at testing if/how PKA regulates Pak1 kinase activity during glucose starvation and refeeding. These two possibilities for the role of Pak1 at RNP granules are not mutually exclusive. It seems likely that these sites could both store Pak1 while keeping it close to a key substrate for rapid regulation in response to glucose. Given that the formation and dissolution of RNP granules is highly regulated by protein kinases and by stress conditions, the regulated localization of protein kinases to these sites may generally contribute to rapid control of RNP granules.

The localization of Pak1 to RNP granules does not appear to involve its canonical ligands within the Cdc42 regulatory module. We did not observe localization of Scd1, Scd2, or Cdc42 at granules. This result is consistent with Pak1 functioning to regulate cell polarity through a new independent mechanism with Sts5, but it also raises questions about how Pak1 is activated at granules. Like other PAKs, Pak1 is thought to adopt an autoinhibited conformation that is relieved by direct binding to activated Cdc42 (*Tu and Wigler, 1999*). While Cdc42 may transiently activate Pak1 at granules, several lines of evidence indicate that Pak1 could be active at granules without Cdc42. First, we and other groups have shown that recombinant full-length Pak1 expressed and purified from bacteria displays strong kinase activity in vitro (*Das et al., 2012*; *Loo and Balasubramanian, 2008*; *Magliozzi al., 2020*; *Pollard et al., 2017*; *Yang et al., 1999*), which means that it is not autoinhibited under these conditions. Second, PAKs can also be activated by various classes of lipids and proteins (*Bokoch et al., 1998*; *Strochlic et al., 2010*; *Zenke et al., 1999*). The proteins that activate PAKs include protein kinases such as PDK, Akt, and CK2 (*King et al., 2000*; *Shin et al., 2013*; *Tang et al., 2000*; *Zhou et al., 2003*), as well as SH3 domain-containing proteins (*Parrini, 2012*). In fission yeast, the SH3 domain protein Skb5 has been shown to enhance Pak1 activity in the absence of Cdc42 in vitro (*Yang*

*et al., 1999*). We do not rule out a possible role for Cdc42 in the function of Pak1 at RNP granules, but these examples show that Pak1 could act at these sites without canonical GTPase-dependent activation.

## Conclusions

Our study provides a molecular link between cell polarity signaling and the regulated assembly of RNP granules. The underlying mechanism involves two conserved protein kinases acting on a shared substrate to control its recruitment into biomolecular condensates. We anticipate that the phase separation properties of many disordered proteins in cells will follow similar regulation by multiple kinases, which facilitates combinatorial responses. Our results also unite two conserved protein kinases in a shared mechanism, raising the possibility for similar connections between PAK and NDR kinases in other organisms. Finally, recruitment of Pak1 and other kinases to RNP granules during stress points to growing roles for these enzymes in controlling the dynamics of granule assembly, disassembly, and function in additional cell types and organisms.

# Materials and methods
## Strain construction and media

Standard *S. pombe* media and methods were used (*Moreno et al., 1991*). Strains used in this study are listed in *Supplementary file 1*. Strains are available upon request. Chromosomal tagging and deletion was performed by PCR-based homologous recombination (*Bähler et al., 1998*). *Psts5-sts5-mNeonGreen-Tadh1* was generated by PCR from genomic DNA of strain JM5505 and inserted into pDC99 by restriction digest and ligation. *Psts5-sts5-9gly5FLAG* was generated by PCR from genomic DNA of strain JM5522 and inserted into pDC99 by restriction digest and ligation. The non-phosphorylatable *sts5* sequences were generated by either site-directed mutagenesis using QuikChange II mutagenesis (Stratagene) or mutant sequences were synthesized as a gBlocks Gene Fragment (Integrated DNA Technologies) and inserted into PCR-linearized fragment from pDC99-*Psts5-sts5-mNeonGreen-Tadh1* or pDC99-*Psts5-sts5-9gly5FLAG-Tadh1* plasmid by repliQa HiFi Assembly Mix (Quantabio). Both strategies to generate point mutations were performed according to the manufacturers' protocols. All plasmids were sequenced by Sanger sequencing for verification. All pDC99 plasmids were linearized by NotI restriction digest and transformed into the *leu1* locus of JM6332 (*sts5Δ ura4-D18*). For RNA extraction, cells were grown in YE4S at 25°C to mid-log phase before harvesting. For growth assays, cells were grown in YE4S to mid-log phase and cells were spotted by 10-fold serial dilutions on YE4S media plates containing indicated compounds at 25, 32, or 37°C for 3–5 days before scanning.

## Protein purification and in vitro kinase assays

Full-length Pak1 constructs (wildtype and catalytically inactive) and full-length Sts5 constructs (wildtype and phosphomutant) were cloned into pGEX6P1 (GST tag) vector (GE Healthcare) for expression in BL21 DE3 *Escherichia coli*. Transformants were grown to log phase at 37°C and then switched to 16°C for 30 min. Protein expression was induced by addition of 1-thio-β-galactopyranoside to a final concentration of 200 μM and cells were incubated overnight at 16°C. Cells were then harvested by centrifugation and lysed three times by French press in lysis buffer (1× PBS, 300 mM NaCl, 200 mM EDTA, 1 mM DTT, and complete EDTA-free protease inhibitor tablets (1 tablet/50 ml buffer); Roche). Triton-X 100 was added to cell lysate to a 1% W/V final concentration for 10 min on ice. Cell lysates were then clarified by centrifugation for 20 min at 14,000 × g at 4°C in a Sorval SS-34 fixed angle rotor (Thermo Scientific). Lysates were then incubated with glutathione-agarose resin (Sigma-Aldrich) for 2 hr rotating at 4°C. Purified proteins were either released from resin by overnight incubation with 3C protease at 4°C or by elution with 20 mM glutathione (pH 8.0).

In vitro kinase assays were performed by incubating purified proteins in kinase assay buffer (50 mM Tris-HCl, pH 7.5, 150 mM NaCl, 10 mM MgCl$_2$, and 1 mM MnCl$_2$) supplemented with 10 μM ATP and 2 μCi γ-$^{32}$P-ATP (blu002z250uc; Perkin Elmer) in 15 μl reactions. Reactions were incubated at 30°C for 30 min and quenched by adding SDS-PAGE sample buffer (65 mM Tris pH 6.8, 3% SDS, 10% glycerol, and 10% 2-mercaptoethanol) and boiling. Gels were dried for 1 hr, and the signal of γ-$^{32}$P-ATP was detected using a PhosphorImager scanned by Typhoon 8600 Variable Mode Imager (GE Healthcare).

Bands on the autoradiograms correspond to the molecular weight where Coomassie-stained protein migrates (*Figure 1—figure supplement 1*).

## Widefield microscopy and analysis

Cells were imaged in either YE4S, EMM4S, or EMM at 25, 32, or 42°C using a DeltaVision imaging system (Applied Precision Ltd.) composed of an IX-inverted widefield microscope (Olympus) with 100× and 60× oil objectives, a CoolSnap HQ2 camera (Photometrics), and an Insight solid-state illumination unit (Applied Precision Ltd.). Images shown as Z-stacks in *Figure 2—figure supplement 2A*, *Figure 5F*, and *Figure 5—figure supplement 1D* were acquired and processed by iterative deconvolution using Soft-WoRx software (Applied Precision Ltd.). For *Figure 1E*, all strains were grown in YE4S at 32°C and treated with 30 μM 3-Brb-PP1 (3-[(3-bromophenyl)methyl]–1-(1,1-dimethylethyl)–1H-pyrazolo[3,4-d]pyrimidin-4-amine) (Abcam) for 1 hr and fixed in ice-cold 70% ethanol. Fixed cells were stored at 4°C for no longer than 24 hr, and cells were stained with Blankophor before imaging to measure septation index. Septation index quantification in *Figure 2G* was performed for the indicated strains grown in EMM4S at 32°C, and cells were stained with Blankophor before imaging. Cell polarity quantification in *Figure 2—figure supplement 2B* was judged by observing actin patch intensity marked by Fim1-mCherry in the indicated strains. Cells with a greater number of patches at one cell end versus the other were considered monopolar. For *Figure 2E*, cells were grown in EMM4S at 32°C and stained with Blankophor before imaging and non-deconvolved single focal plane images are shown with an inverted lookup table. Quantification of cell width in *Figure 2F* and *Figure 8—figure supplement 1B* was measured by drawing a line across the division plane in septating cells stained by Blankophor. Quantification of cell length in *Figure 8—figure supplement 1B* was measured by drawing a line over the entire cell length in septating cells stained by Blankophor. For *Figure 5A* and *Figure 5—figure supplement 3A*, cells were grown in EMM4S 2% glucose to mid-log phase and spun down and washed 4× in EMM4S 0% glucose, resuspended and incubated in EMM4S 0% glucose shaking at 25°C for 60 min before imaging. Single-channel fluorescence non-deconvolved single focal plane images are shown with an inverted lookup table. For *Figure 5F* and *Figure 5—figure supplement 1D*, -Nitrogen cells were grown in EMM and switched to EMM lacking nitrogen source overnight at 25°C before imaging; 1 M KCl stress occurred for 30 min in EMM4S + 1 M KCl 25°C before imaging; and heat shock stress was performed with cells grown in EMM4S at 25°C and then switched to 42°C for 20 min before imaging. All image analysis was performed on ImageJ2 (National Institutes of Health). Statistical differences were assessed by one-way ANOVA or Welch's *t* test using GraphPad.

## Spinning disc microscopy and analysis

Images for *Figure 3A* were taken with a spinning disc confocal system (Micro Video Instruments) featuring a Nikon Eclipse Ti base equipped with an Andor CSU-W1 two-camera spinning disc module, dual Zyla sCMOS cameras (Andor), an Andor ILE laser module, and a Nikon 100 x Plan Apo $\lambda$ 1.45 oil immersion objective. Images shown are maximum intensity projections of the top half of cells. Colocalization analysis, number of Sts5 granules, and number of P bodies in *Figure 3B* were performed using ImageJ2 plugin ComDet v.0.5.0. Square ROIs of with height = 22.23 μm and width = 26.52 μm were used for analysis. Channel 1 was set to 3.00 pixels and a threshold of 12. Channel 2 was set to 4.00 pixels and a threshold of 12. Both channels were set to include larger particles and segment larger particles. For calculating colocalization, maximum distance between colocalized spots was set to 3.00 pixels and ROI shapes were set for ovals.

All other spinning disc confocal microscopy was performed using a Nikon Eclipse Ti-E microscope stand with a Yokogawa, two-camera, CSU-W1 spinning disc system with a Nikon LU-N4 laser launch. This system has Photometrics Prime BSI sCMOS cameras. For *Figures 1D, H , and 2D*, Sts5-mNG granules quantification was done using ImageJ2 plugin Trackmate using a 0.4 μm size and 10 threshold setting for each strain using maximum intensity projections of the whole cell. Spinning disc images showed indicated protein localization in 0% glucose, and cells were grown in EMM4S 2% glucose to mid-log phase and spun down and washed 4x in EMM4S 0% glucose, resuspended, and incubated in EMM4S 0% glucose shaking at room temperature for 40 min before imaging. Sts5-mNG stress granule dissolution quantification was done by washing cells in EMM4S 0% glucose media 4×, and cells were resuspended in EMM4S 0% glucose and incubated shaking at 25°C for 40 min. After 40 min, timepoint 0 in stress granule dissolution sample was spun down and fixed in ice-cold 70% ethanol

and stored at 4°C until imaging. For the timepoints following timepoint 0, cells in EMM4S 0% glucose were spun down and washed into EMM4S 2% glucose, incubated shaking at 25°C, and samples were removed at the indicated timepoints and fixed in ice-cold 70% ethanol and stored at 4°C until imaging. For *Figure 8C* and *Figure 8—figure supplement 1C,D*, background subtracted integrated densities of clustered Sts5-mNG signal were measured using circular ROIs drawn around the clustered signal from sum projections of the whole cell. For each timepoint, 50 clusters were measured and plotted. For *Figure 8D*, quantification was done using ImageJ2 plugin Trackmate using a 0.6 µm size and 60 threshold setting for each strain using sum intensity projections of the whole cell. Statistical differences were assessed by one-way ANOVA or Welch's *t* test as described below using GraphPad.

## Co-immunoprecipitation, western blotting, and analysis

Co-immunoprecipitation experiments in *Figure 3C* and *Figure 3—figure supplement 1* were performed with a modified protocol from *Deng and Moseley, 2013*. Briefly, 12 $OD_{595}$ cells of indicated strains were grown and harvested, washed 1× in milliQ water, snap frozen in liquid nitrogen, and stored at –80°C. Cell lysates were generated by lysing cells in 200 µl lysis buffer (20 mM HEPES, 150 mM NaCl, 1 mM EDTA, 0.2% TX-100, 50 mM NaF, 50 mM b- glycerolphosphate, 1 mM sodium orthovanadate, 1 mM PMSF, and one protease inhibitor cocktail tablet) with acid-washed glass beads (Sigma) in a Mini-BeadBeater-16 (BioSpec) for two rounds of 1 min cycles with 2 min on ice between cycles in a cold room. Clarified lysates were generated by centrifugation at 14,000 × *g* for 5 min. Clarified lysates were incubated with anti-GFP agarose resin (courtesy of Dartmouth Bio-MT facility) that was washed 3× in lysis buffer prior to lysate addition. Beads and lysate were incubated rotating for 60 min at 4°C, washed 5× in lysis buffer, resuspended in SDS-PAGE sample buffer (described below), boiled for 5 min at 99°C, followed immediately by SDS-PAGE and western blotting. Western blots were probed with α-GFP (*Moseley et al., 2009*) and α-FLAG M2 (Sigma) antibodies. Western blots were developed using an Odyssey CLx Imaging System (LI-COR). For quantification of Sts5-FLAG protein levels in *Figure 3D*, a rectangular ROI was drawn around each band. Mean intensity was measured, and background was subtracted for both FLAG and GFP IP signals using Image Studio Lite (LI-COR). The ratio of Sts5-FLAG to Rad24-GFP IP signal was normalized to the wildtype and calculated for three biological replicates of each strain indicated. Statistical differences were assessed by one-way ANOVA using GraphPad.

For western blots in *Figure 2B*, *Figure 2—figure supplement 1*, and *Figure 4A and D*, samples of two $OD_{595}$ mid-log phase cells were harvested and snap-frozen in liquid nitrogen. Whole-cell extracts were generated by lysing cells in 3× sample buffer (65 mM Tris, pH 6.8, 3% SDS, 10% glycerol, 10% 2-mercaptoethanol, 50 mM NaF, 50 mM β-glycerolphosphate, 1 mM sodium orthovanadate, and protease inhibitor cocktail) with acid-washed glass beads (Sigma) in a Mini-BeadBeater-16 (BioSpec) for 2 min in a cold room. Samples were then heated to 99°C for 5 min, briefly centrifuged to pellet insoluble material, and the supernatant was used as whole-cell extract. Western blots were probed with α-HA (Covance), α-Cdc2 (SC-53217), and α-FLAG M2 (Sigma). Western blots were developed using an Odyssey CLx Imaging System (LI-COR). For quantification of protein levels in *Figure 4B, E*, a rectangular ROI was drawn around each band. Mean intensity was measured, and background was subtracted for both HA and Cdc2 loading control signals using Image Studio Lite (LI-COR). The ratio of HA to Cdc2 signal was calculated for three biological replicates of each strain indicated. Statistical differences were assessed by one-way ANOVA using GraphPad. For *Figure 2B* and *Figure 2—figure supplement 1*, samples were run on a 6% SDS-PAGE gel containing 100 µM Phosbind (Apexbio) according to the manufacturer's protocol to separate phosphorylated isoforms of Sts5. Membranes were treated with Ponceau stain and scanned to display total protein loaded in each lane.

Raw unedited versions of all gels and western blots are provided as source data files. Note that all blots imaged on Odyssey CLx Imaging System (LI-COR) were scanned 'upside-down' and therefore were rotated 180° prior to formatting for figures.

## RNA extraction and NanoString analysis

RNA extraction for wildtype and Sts5-2A cells was performed using MasterPure Yeast RNA Purification Kit (Epicentre) according to the manufacturer's protocol. The NanoString nCounter Elements XT assay was performed according to the manufacturer's instructions (NanoString Technologies, Seattle, WA) at the Molecular Biology Core Facility at Dartmouth College. Total RNA was hybridized with

user-designed target-specific oligonucleotide probes and TagSets, which can detect gene expression levels by using fluorescently labeled barcoded reporters that recognize sequence-specific tags. Reactions were incubated at 67°C for 18 hr. Subsequently, samples were loaded to the nCounter PrepStation (NanoString Technologies), which automatically performs purification steps and cartridge preparation. Finally, the cartridges containing immobilized and aligned reporter complexes were transferred to the nCounter Digital Analyzer (NanoString Technologies), set at a high-resolution setting, which captures up to 280 fields of view (FOVs) per sample providing all gene counts. Raw data was collected and preprocessed by nSolver Analysis Software v4.0 (NanoString Technologies). For data analysis, a two-step normalization method was applied as described (*Doing et al., 2020*). Transcript counts were normalized to the geometric mean of positive control samples and three housekeeping genes (Act1, Cdc2, and Nda3). Probe sets and raw transcript count data are listed in *Supplementary files 2 and 3*, respectively.

## Statistical analyses

Welch's *t* test was used to assess differences for *Figures 1D, 3B, 4C and F*, *Figure 4—figure supplement 1A,B*, *Figure 8C and D*, and *Figure 8—figure supplement 1B-D*. One-way ANOVA followed by Dunnett's multiple comparison test was used to assess differences for *Figure 2F, G* and *Figure 4B, E*. One-way ANOVA followed by Tukey's multiple comparison test was used to assess differences for *Figures 1E, H and 2D*, *Figure 2—figure supplement 2B*, and *Figure 3D*. GraphPad Prism 7 was used for all statistical analyses. Dunnett's tests were used when comparing every mean within an experiment to a control mean, while Tukey's tests were used when comparing every mean within an experiment to each other.

## Acknowledgements

We thank the members of the Moseley lab, Corey Allard, and Erik Griffin for comments on the manuscript. We thank the Biomolecular Targeting Core (BioMT) (P20-GM113132) and the Imaging Facility at Dartmouth for equipment and technical assistance, Christian Lytle and Deborah Hogan for help in performing and analyzing NanoString experiments, and Sophie Martin and Rosa Aligue for sharing strains. This work was supported by grants from the NIH (R01-GM099774 and R01-GM133856) to JBM.

## Additional information

### Funding

| Funder | Grant reference number | Author |
|---|---|---|
| National Institute of General Medical Sciences | R01-GM099774 | James B Moseley |
| National Institute of General Medical Sciences | R01-GM133856 | James B Moseley |
| National Institute of General Medical Sciences | P20-GM113132 | Joseph O Magliozzi James B Moseley |

The funders had no role in study design, data collection and interpretation, or the decision to submit the work for publication.

### Author contributions

Joseph O Magliozzi, Conceptualization, Formal analysis, Investigation, Methodology, Writing - original draft, Writing - review and editing; James B Moseley, Conceptualization, Formal analysis, Funding acquisition, Methodology, Project administration, Supervision, Writing - original draft, Writing - review and editing

### Author ORCIDs

Joseph O Magliozzi (iD) http://orcid.org/0000-0002-4173-2369
James B Moseley (iD) http://orcid.org/0000-0002-7354-7416

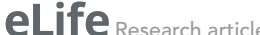 Research article

Cell Biology

**Decision letter and Author response**
Decision letter https://doi.org/10.7554/eLife.67648.sa1
Author response https://doi.org/10.7554/eLife.67648.sa2

## Additional files

### Supplementary files
- Supplementary file 1. Table of yeast strains and plasmids used in this study.
- Supplementary file 2. Table of NanoString probe names and sequences.
- Supplementary file 3. Table of data from NanoString gene expression analysis.
- Transparent reporting form

### Data availability
All relevant data are included in the manuscript and supporting files.

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
