## [Decision Letter]

**Acceptance summary:**

This paper uses fission yeast as. a model to understand how RNP granules participate in important signal transduction events and how these control cell physiology, via growth/polarity control. The key molecule investigated is the highly conserved Pak1 kinase (well characterized as a Cdc42 GTPase effector). The paper brings together investigation of sts5, pak1-kinase and how these together regulate P bodies during glucose starvation. The work is of broad interest to cell biologists and RNA biologists working across scales.

**Decision letter after peer review:**

Thank you for submitting your article "Pak1 kinase regulates ribonucleoprotein granules through Sts5 for polarized growth and the glucose starvation response" for consideration by *eLife*. Your article has been reviewed by 3 peer reviewers, and the evaluation has been overseen by a Reviewing Editor and Anna Akhmanova as the Senior Editor. The following individuals involved in review of your submission have agreed to reveal their identity: Fulvia Verde (Reviewer #2); Dr. Takashi Toda (Reviewer #3).

Please make sure you address the following essential revisions in full, some of which require new experiments.

Essential revisions:

1) The usage of ribonucleoprotein (RNP) granules

In this manuscript, two types of RNP granules are investigated: one is P body and the other is stress granule (SG). P bodies constitutively exist under all conditions, while SGs assemble only under stressed conditions including glucose starvation. It is true that P bodies coalesce with SGs under stresses, but as far as I am concerned, these two types of RNP granules are different. I think it would be better for the authors to use SGs in the text instead of RNP granules when they describe results under glucose deprivation (Figures 4-7). Also, in Figure 7D, RNP should be replaced with SG.

2) Binding between Sts5 and 14-3-3

Previous work (Nuñez et al., 2016) showed that Orb6-dependent phosphorylation of Sts5 promotes binding between Sts5 and 14-3-3, thereby inhibiting its localization to P bodies. Is this also the case for Pak1-dependent phosphorylation of Sts5? Or in this case, does phosphorylation of serines (S261, S264) within the intrinsically disordered region (IDR) somehow (maybe by disrupting the disordered feature of the IDR?) interfere with the inclusion of Sts5 into P bodies?

3) A marker of SGs

This is related to point (1). I think that the authors should have used an authentic marker of SGs, not Sum2 (a marker of P bodies), in experiments under glucose deprivation. At least, co-localization between Sum2 and a marker for SGs under this condition should be shown.

4) Dissolution of Sts5 from RNP granules upon glucose re-addition: Figure 7

Tethering experiments presented in Figure 7 are of great interest. The difference with regards to the degree of dissolution of Sts5 between wild type cells and those containing CAAX-tethered Pak1 is modest, yet I think it convincing after I scrutinize all the data including those shown in Supplements. I am interested in the physiological consequences, if any, by this tethering. It looks to this referee that cell shape is substantially different between wild type and Pak1-CAAX strains (shown in the bottom row in Figure 7B). I would like to request showing quantification of cell length and width in these cells. I suspect that cells containing Pak1-CAAX display more roundish morphology upon glucose re-addition, which would represent down-regulation of Sts5 function.

5) Does loss of sts5 suppresses any viability or cell separation defects of pak1 mutants, similarly to what happens in orb6 mutants?

6) Following Pak1 kinase activity during glucose deprivation and refeeding could shed light on the dynamics of the Pka1/Pak1 regulatory pathway. Does Pka1 affects Pak1 kinase activity with or without glucose stress and during refeeding?

7) Do pak1-CAAX mutants display a delay in resuming polarized cell growth following glucose starvation, as compared with controls? This effect would further support a role for Pak1 in Sts5 dissolution and resumption of translation of specific mRNAs.

(8) The claim that absence of Orb6 and Pak1-dependent phosphorylation of Sts5 concentrates Sts5 predominantly to P bodies and causes downregulation of translation of target mRNA should be reconsidered. While Figure 3B shows colocalization among some Sts5(2A)+ and Sum2+ puncta, some Sts5(2A)+ Sum2- and Sts5(2A)- Sum2+ puncta are also present. The effect on translation levels should be validated by investigating at least a few other Sts5 mRNA targets (both the levels of mRNA and their protein products should be tested). The model presented in Figure 3F seems to be premature at this stage.

9) The difference in the rates of Sts5 puncta dissolution in presence or absence of punctate Pak1 (Figure 7C) is modest. The authors may consider investigating the physiological consequence of the observed difference in dissolution rates, which remains unclear. Further, not all Sts5+ puncta contain Pak1 (Figure 4D). Have the authors considered scoring the dissolution rates of Sts5+ Pak1+ vs. Sts5+ Pak1- puncta separately?

*Reviewer #1:*

Magliozzi et al. study the interaction of three proteins that have earlier been identified to control cell shape in fission yeast. Two of these are the kinases Pak1 and Orb6. The third is an RNA-binding protein called Sts5. It has been shown earlier that Sts5 uses an intrinsically disordered region (IDR) to concentrate in P bodies, and Orb6-dependent phosphorylation of Sts5-IDR reduces the association of Sts5 with P bodies.

It was unknown if Sts5 is also phosphorylated by Pak1. The authors begin by using biochemistry to show that Pak1 phosphorylates Sts5-IDR at residues S261 and S264. Orb6 has been shown earlier to phosphorylate Sts5 at residue S86. Using non-phosphorylation-competent mutants of Sts5 in vivo, the authors demonstrate that phosphorylation at these residues is important to maintain diffuse distribution of Sts5 in the cytoplasm. In absence of phosphorylation by Orb6 and/or Pak1, increasing fractions of Sts5 protein concentrate in cytoplasmic puncta. This is accompanied by defects in cell morphology, septation and viability. The authors claim that these Sts5+ puncta are P bodies, based on (a) colocalization of some of these puncta with the P body marker Sum2, and (b) decrease of cellular levels of a protein SSp1 concomitant with concentration of Sts5 in puncta (Sts5 is known to bind ssp1 mRNA).

Next, the authors show that the intracellular localization of Pak1 depends on the amount of glucose in the growth medium (2% vs 0% glucose). In absence of glucose, Pak1 relocates from the cell cortex to join Sts5 in P bodies (tracked by Sum2). This relocalization is dependent on Pka1 (the catalytic subunit of Protein kinase A) in contrast to the Cdc42-associated cell polarity regulatory module. In contrast to Pak1, absence of glucose in the medium did not have any effect on the localization of the other kinase Orb6. In order to address the consequence of relocalization of the kinase Pak1 to P bodies, the authors generated a construct Pak1-CAAX that remained localized to the cortex in absence of glucose. The authors claim that localization of Pak1 to P bodies increase the rate of dissolution of Sts5 puncta.

Taken together, the authors provide new mechanistic insights into the interplay of Pak1, Orb6 and Sts5 when fission yeast is grown in medium containing different amounts of glucose.

(1) The claim that absence of Orb6 and Pak1-dependent phosphorylation of Sts5 concentrates Sts5 predominantly to P bodies and causes downregulation of translation of target mRNA would need to be reconsidered. While Figure 3B shows colocalization among some Sts5(2A)+ and Sum2+ puncta, some Sts5(2A)+ Sum2- and Sts5(2A)- Sum2+ puncta are also present. The effect on translation levels should be validated by investigating at least a few other Sts5 mRNA targets (both the levels of mRNA and their protein products should be tested). The model presented in Figure 3F seems to be premature at this stage.

2) The difference in the rates of Sts5 puncta dissolution in presence or absence of punctate Pak1 (Figure 7C) is modest. The authors may consider investigating the physiological consequence of the observed difference in dissolution rates, which remains unclear. Further, not all Sts5+ puncta contain Pak1 (Figure 4D). It would be good to know whether the authors have considered scoring the dissolution rates of Sts5+ Pak1+ vs. Sts5+ Pak1- puncta separately.

3) The authors could consider revising the Introduction and Discussion section to make them more focused.

4) The cell outlines in Figure 3B (Sum2-tdTomato panel) are misleading.

*Reviewer #2:*

In this paper, the authors characterize the role of fission yeast Pak1 kinase in the control of mRNA binding protein Sts5 and in P-bodies assembly. Sts5 was previously identified as a substrate of Pak1 kinase in a previous paper by the same group. In this current manuscript the authors characterize the role of Pak1-dependent phosphorylation of two sites (S261 and S264), in the unstructured domain of Sts5, and show that loss of phosphorylation promotes Sts5 assembly. A sts5 mutant mimicking a dephosphorylated state of these two sites presents increased asymmetric cell growth (monopolarity) and increased septation index.

Interestingly, increased Sts5 assembly and morphological anomalies comprise a phenotype that is very reminiscent of the Sts5-S86A mutant, which is modified on a residue that is phosphorylated by Orb6 kinase and was previously characterized by a different lab. Consistent with previous observations that orb6 and pak1 mutants are synthetically lethal, the S261A mutation displays an additive effect with S86A, in promoting Sts5 granule formation, altering cell morphology, and delaying septation.

Using the Sum2 protein as a P-bodies marker, the authors show that increased granule assembly of the Sts5 S86A/S261A protein leads to increased numbers of P-bodies; further, this effect leads to decreased Ssp1 protein levels of Ssp1, a protein encoded by one of the Sts5-associated mRNAs, also consistent with previous findings.

Next, the authors define a crucial difference between Orb6 and Pak1, in that during glucose deprivation, a stress that leads to P bodies assembly, Pak1 kinase localizes to P bodies (and co-localizes with Sts5), while Orb6 kinase does not. Pak1 localization to P-bodies does not include factors that cooperate with Pak1 in the control of cell polarity, such as Scd1, Scd2, or Cdc42 GTPase. The authors note that other kinases, such as Kin1 and Pck2, also have the ability to associate into granules that, in part, include Sum2. Finally, they find that Pak1 localization to P bodies during stress depends on the glucose sensing pathway mediated by Pka1 kinase. Consistent with a potential role for Pak1 kinase in Sts5 granule dissolution, preventing Pak1 localization to the cytoplasm subtly delays Sts5 granule disassembly upon glucose refeeding.

This paper highlights a novel function for Pak1 kinase in the control of RNP dynamics, and nicely align itself with previously published data. It extends our knowledge of the signal integration that controls the coalescence of Sts5 into granules and the assembly of P-bodies. Further, these observations also identify a role for glucose sensing Pka1 kinase in the control of Pak1 localization to the P-bodies, a function that is independent of the polarity factors Pak1 is known to associate with.

1. It should be noted that observations regarding Orb6 kinase phosphorylation of Serine 86, and the role of this residue in regulating Sts5 granule assembly were previously published (Chen et al., 2019). The role of Pak1 in Sts5 phosphorylation, and the specific residues phosphorylated by Pak1 kinase were also previously identified (Magliozzi et al., 2020). Therefore, the fact that Orb6 and Pak1 phosphorylate distinct residues of Sts5 or that Orb6 regulates Sts5 IDR is not a novel discovery in this paper (as mentioned repeatedly in lines 83-84 , 99-100, 158-161, 177-78, 186-87, etc.). However, the fact that Pak1 function had additive properties to Orb6 kinase in controlling the state of Sts5 granule assembly, in particular under conditions of glucose deprivation, is a novel, interesting expansion of knowledge.

2. Importantly, the authors refer to "stress granules" in several titles (line 185, line 225, line 244). These statements do not correspond to the data presented and should be corrected. Stress granules (SG granules) are separate membraneless organelles that contain specific factors, differentiating them from P-bodies. The marker Sum2, which is used in the paper, does not identify stress granules, but rather it colocalizes with Dcp1, a component of P-bodies. Therefore, data presented here supports the role of Pak1 in the control of P-bodies, not stress granules.

3. The 2% glucose control cells used for experiments shown in Figure 4 (Figure 4, Supplement 1) appear very stressed. P-bodies (as visualized by Sum2) do not condense in healthy, exponentially growing cells. While this effect does not invalidate the results of the experiments shown in Figure 4, it would need to be corrected.

4. Extending the quantification in Figure 7B, to include numbers of Sts5 granules (not only overall fluorescence intensity) would give a more precise assessment of the effects of Pak1 removal on Sts5 granule disassembly.

The overall results are interesting and the experiments are solidly performed and analyzed.

The paper could be strengthened by genetic epistasis experiments and by following up on the novelty of the Pka1-Pak1 regulatory axis.

Comments for the authors:

1. Does loss of sts5 suppresses any viability or cell separation defects of pak1 mutants, similarly to what happens in orb6 mutants?

2. Following Pak1 kinase activity during glucose deprivation and refeeding could shed light on the dynamics of the Pka1/Pak1 regulatory pathway. Does Pka1 affects Pak1 kinase activity with or without glucose stress and during refeeding?

3. Do pak1-CAAX mutants display a delay in resuming polarized cell growth following glucose starvation, as compared with controls? This effect would further support a role for Pak1 in Sts5 dissolution and resumption of translation of specific mRNAs.

4. What is the rationale of switching between Tukey's and Dunnett's methods in some of the ANOVA analyses?

*Reviewer #3:*

Fission yeast RNA-binding protein Sts5 is known to localize to P bodies (intracellular granules consisting of RNAs and proteins (ribonucleoproteins, RNPs)), by which this protein plays an important role in cell morphogenesis and polarity. P bodies-localizing Sts5 binds many species of mRNAs encoding regulatory factors required for polarized cell growth, thereby repressing their translation. The Orb6 kinase (a member of the NDR/LATS kinase family) directly phosphorylates Sts5, which restrains localization of Sts5 to P bodies, resulting in proper polarized growth.

In this manuscript, Magliozzi and Moseley have shown that Sts5 is phosphorylated also by the Pak1 kinase (the p21-activated kinase). The phosphorylation site is different from that mediated by Orb6. Interestingly, this phosphorylation also inhibits P bodies-localization of Sts5, which acts independently of and additively with Orb6. The authors have found that upon glucose deprivation, Pak1 together with Sts5 is recruited to RNP granules (stress granules, SGs). Localization of Pak1, but not that of Sts5, is dependent upon protein A kinase (PKA) signalling. They propose that Pak1 localization to RNP granules promotes rapid dissolution of Sts5 from SGs upon glucose re-addition.

Strengths:

A technically sound dataset is presented. Results are clearly and logically shown.

Weaknesses:

The usage of "RNP granule' is confusing: it sometimes means P body, while in other cases (experiments under glucose deprivation), it stands for SG.

A mechanism by which Pak1-dependent phosphorylation of Sts5 inhibits Sts5 localization to P-bodies remains unknown.

A mechanism by which PKA signalling promotes Pak1 localization to SGs has not been explored.

Comments for the authors:

1) The usage of ribonucleoprotein (RNP) granules

In this manuscript, two types of RNP granules are investigated: one is P body and the other is stress granule (SG). P bodies constitutively exist under all conditions, while SGs assemble only under stressed conditions including glucose starvation. It is true that P bodies coalesce with SGs under stresses, but as far as I am concerned, these two types of RNP granules are different. I think it would be better for the authors to use SGs in the text instead of RNP granules when they describe results under glucose deprivation (Figures 4-7). Also, in Figure 7D, RNP should be replaced with SG.

2) Binding between Sts5 and 14-3-3

Previous work (Nuñez et al., 2016) showed that Orb6-dependent phosphorylation of Sts5 promotes binding between Sts5 and 14-3-3, thereby inhibiting its localization to P bodies. Is this also the case for Pak1-dependent phosphorylation of Sts5? Or in this case, does phosphorylation of serines (S261, S264) within the intrinsically disordered region (IDR) somehow (maybe by disrupting the disordered feature of the IDR?) interfere with the inclusion of Sts5 into P bodies?

3) A marker of SGs

This is related to point (1). I think that the authors should have used an authentic marker of SGs, not Sum2 (a marker of P bodies), in experiments under glucose deprivation. At least, co-localization between Sum2 and a marker for SGs under this condition should be shown.

4) Dissolution of Sts5 from RNP granules upon glucose re-addition: Figure 7

Tethering experiments presented in Figure 7 are of great interest. The difference with regards to the degree of dissolution of Sts5 between wild type cells and those containing CAAX-tethered Pak1 is modest, yet I think it convincing after I scrutinize all the data including those shown in Supplements. I am interested in the physiological consequences, if any, by this tethering. It looks to this referee that cell shape is substantially different between wild type and Pak1-CAAX strains (shown in the bottom row in Figure 7B). I would like to request showing quantification of cell length and width in these cells. I suspect that cells containing Pak1-CAAX display more roundish morphology upon glucose re-addition, which would represent down-regulation of Sts5 function.

---

## [Author Response]

Essential revisions:1) The usage of ribonucleoprotein (RNP) granulesIn this manuscript, two types of RNP granules are investigated: one is P body and the other is stress granule (SG). P bodies constitutively exist under all conditions, while SGs assemble only under stressed conditions including glucose starvation. It is true that P bodies coalesce with SGs under stresses, but as far as I am concerned, these two types of RNP granules are different. I think it would be better for the authors to use SGs in the text instead of RNP granules when they describe results under glucose deprivation (Figures 4-7). Also, in Figure 7D, RNP should be replaced with SG.

Thanks for this recommendation. We now refer to stress granules for experiments under glucose starvation. This distinction is strengthened by our new data showing colocalization of Pak1 and bona fide stress granule marker Pabp (new Figure 5 Supp 2).

2) Binding between Sts5 and 14-3-3Previous work (Nuñez et al., 2016) showed that Orb6-dependent phosphorylation of Sts5 promotes binding between Sts5 and 14-3-3, thereby inhibiting its localization to P bodies. Is this also the case for Pak1-dependent phosphorylation of Sts5? Or in this case, does phosphorylation of serines (S261, S264) within the intrinsically disordered region (IDR) somehow (maybe by disrupting the disordered feature of the IDR?) interfere with the inclusion of Sts5 into P bodies?

Thanks for this suggestion, which led to new figures in the revised manuscript (Figures3C,D and Figure 3 Supp 1). We tested Sts5-Rad24 interactions using coimmunoprecipitation and found that S261A reduced this interaction similarly to S86A. Further, the double mutant had an additive effect and nearly abolished Sts5-Rad24 interactions. We conclude that Pak1 phosphorylates Sts5 to promote its binding to Rad24, and this interaction prevents protein clustering.

3) A marker of SGsThis is related to point (1). I think that the authors should have used an authentic marker of SGs, not Sum2 (a marker of P bodies), in experiments under glucose deprivation. At least, co-localization between Sum2 and a marker for SGs under this condition should be shown.

We appreciate this recommendation, which led to new data and a clearer picture of Pak1 localization to stress granules. As shown in the new Figure 5 Supp 2, we found that Pak1 and Sts5 both colocalize with Pabp, in addition to their colocalization with Sum2. Pabp is the major poly-A binding protein and a marker of stress granules. It is important to note that Pabp and Sum2 also colocalize with each other during glucose deprivation, consistent with previous studies showing that P bodies and stress granules are overlapping structures in yeast cells (Buchan et al., *J Cell Biol*., 2008; Protter and Parker, *Trends Cell Biol*., 2016).

4) Dissolution of Sts5 from RNP granules upon glucose re-addition: Figure 7Tethering experiments presented in Figure 7 are of great interest. The difference with regards to the degree of dissolution of Sts5 between wild type cells and those containing CAAX-tethered Pak1 is modest, yet I think it convincing after I scrutinize all the data including those shown in Supplements. I am interested in the physiological consequences, if any, by this tethering. It looks to this referee that cell shape is substantially different between wild type and Pak1-CAAX strains (shown in the bottom row in Figure 7B). I would like to request showing quantification of cell length and width in these cells. I suspect that cells containing Pak1-CAAX display more roundish morphology upon glucose re-addition, which would represent down-regulation of Sts5 function.

The reviewer is correct that Pak1-CAAX cells have increased width when compared to wild type cells. We have quantified this effect in the new Figure 8 Supp 1B.

5) Does loss of sts5 suppresses any viability or cell separation defects of pak1 mutants, similarly to what happens in orb6 mutants?

We show that *sts5∆* suppresses cell separation defects of *pak1-as* mutants in Figure 1E. This result supports the model that aberrant Sts5 clusters observed in *pak1-as* cells are toxic structures.

6) Following Pak1 kinase activity during glucose deprivation and refeeding could shed light on the dynamics of the Pka1/Pak1 regulatory pathway. Does Pka1 affects Pak1 kinase activity with or without glucose stress and during refeeding?

We agree with the reviewer, and attempted to address these questions with new experiments. Unfortunately, two technical problems prevented us from obtaining clear answers. First, our preferred method to assay kinase activity (^32^P-ATP) is not currently available to us due to covid-related limitations in access. As an alternative, we attempted to test Pak1 kinase activity with SDS-PAGE band shifts but did not obtain conclusive results. Second, our attempts to isolate Pak1 from cells at timepoints during glucose starvation-refeeding did not yield clean purifications for in vitro testing. We have several ideas for how to address this important question going forward, but clearly it will require an extended period of time to establish reliable assays. To emphasize the importance of the reviewer’s suggestion, we have added the following statement into the revised manuscript: “Future studies will be directed at testing if/how PKA regulates Pak1 kinase activity during glucose starvation and refeeding.”

7) Do pak1-CAAX mutants display a delay in resuming polarized cell growth following glucose starvation, as compared with controls? This effect would further support a role for Pak1 in Sts5 dissolution and resumption of translation of specific mRNAs.

We agree with the reviewer’s suggestion but had difficulty obtaining clear results to this question. In brightfield microscopy videos, we observed a long delay in the resumption of tip growth for Pak1-CAAX versus wild type cells. However, the Pak1-CAAX cells have polarity and tip growth defects even under high glucose conditions, likely because the CAAX motif restricts Pak1 dynamics under all conditions. In our videos, it was not clear if the Pak1-CAAX cells resumed growth isotropically upon glucose addition. In addition, the delay in resuming clearly polarized growth was substantially longer than the delay in Sts5 dissolution, indicating that the growth delay likely reflects myriad changes in this strain. To avoid overinterpreting this result, we would prefer not to include it in this manuscript. We hope to clarify this issue with additional assays and controls in future work.

(8) The claim that absence of Orb6 and Pak1-dependent phosphorylation of Sts5 concentrates Sts5 predominantly to P bodies and causes downregulation of translation of target mRNA should be reconsidered. While Figure 3B shows colocalization among some Sts5(2A)+ and Sum2+ puncta, some Sts5(2A)+ Sum2- and Sts5(2A)- Sum2+ puncta are also present. The effect on translation levels should be validated by investigating at least a few other Sts5 mRNA targets (both the levels of mRNA and their protein products should be tested). The model presented in Figure 3F seems to be premature at this stage.

We extended our results beyond Ssp1 in response to this suggestion. In Figure 4 of the revised manuscript, we report that Cmk2 protein levels are also reduced in the *sts5-2A* mutant, similarly to Ssp1. The transcript levels of both *ssp1* and *cmk2* were previously shown to be upregulated in *sts5∆* cells (Nunez et al., 2016), consistent with lower protein levels in the hyperactive *sts5-2A* mutant. In addition, we performed Nanostring experiments to measure mRNA levels of Sts5-regulated transcripts and observed reduced levels for *ssp1*, *cmk2*, *psu1*, and *efc1* in the *sts5-2A* mutant. These new results are shown in Figures 4C-F as well as Figure 4 Supplement 1. We have also updated the model (now Figure 4G) to reflect reduced Cmk2 levels in *sts5-2A* mutant cells.

9) The difference in the rates of Sts5 puncta dissolution in presence or absence of punctate Pak1 (Figure 7C) is modest. The authors may consider investigating the physiological consequence of the observed difference in dissolution rates, which remains unclear. Further, not all Sts5+ puncta contain Pak1 (Figure 4D). Have the authors considered scoring the dissolution rates of Sts5+ Pak1+ vs. Sts5+ Pak1- puncta separately?

We agree that comparing the composition and dissolution kinetics of individual granules could reveal interesting differences. However, our analysis relied on fixed cells due to technical difficulties including photobleaching during time-lapse experiments. We attempted to address this question using several different approaches including microfluidics and chambered coverslips, but we were unable to reliably track and measure single granules. We will continue to trouble-shoot these issues going forward, but for the revised manuscript we focused on extending our fixed cell approach as in the new Figure 8D.

Reviewer #1:[…] (1) The claim that absence of Orb6 and Pak1-dependent phosphorylation of Sts5 concentrates Sts5 predominantly to P bodies and causes downregulation of translation of target mRNA would need to be reconsidered. While Figure 3B shows colocalization among some Sts5(2A)+ and Sum2+ puncta, some Sts5(2A)+ Sum2- and Sts5(2A)- Sum2+ puncta are also present. The effect on translation levels should be validated by investigating at least a few other Sts5 mRNA targets (both the levels of mRNA and their protein products should be tested). The model presented in Figure 3F seems to be premature at this stage.

Please see our response to essential revision #8.

2) The difference in the rates of Sts5 puncta dissolution in presence or absence of punctate Pak1 (Figure 7C) is modest. The authors may consider investigating the physiological consequence of the observed difference in dissolution rates, which remains unclear. Further, not all Sts5+ puncta contain Pak1 (Figure 4D). It would be good to know whether the authors have considered scoring the dissolution rates of Sts5+ Pak1+ vs. Sts5+ Pak1- puncta separately.

Please see our response to essential revision #9.

3) The authors could consider revising the Introduction and Discussion section to make them more focused.

We have revised both sections with this goal in mind.

4) The cell outlines in Figure 3B (Sum2-tdTomato panel) are misleading.

Thanks for this comment, we have re-drawn the cell outlines as suggested.

Reviewer #2:[…] 1. It should be noted that observations regarding Orb6 kinase phosphorylation of Serine 86, and the role of this residue in regulating Sts5 granule assembly were previously published (Chen et al., 2019). The role of Pak1 in Sts5 phosphorylation, and the specific residues phosphorylated by Pak1 kinase were also previously identified (Magliozzi et al., 2020). Therefore, the fact that Orb6 and Pak1 phosphorylate distinct residues of Sts5 or that Orb6 regulates Sts5 IDR is not a novel discovery in this paper (as mentioned repeatedly in lines 83-84 , 99-100, 158-161, 177-78, 186-87, etc.). However, the fact that Pak1 function had additive properties to Orb6 kinase in controlling the state of Sts5 granule assembly, in particular under conditions of glucose deprivation, is a novel, interesting expansion of knowledge.

We have revised the text to focus on the additive nature of this regulatory mechanism as suggested by the reviewer. It is worth noting that Figure 1 of our current manuscript is dedicated to establishing Sts5-S261 as a direct target of Pak1, which was not established in our 2020 publication and therefore represents a novel finding. Based on this novel finding which required in vitro reconstitution and phosphorylation-site mutants (Figure 1), we can now conclude that Pak1 and Orb6 phosphorylate distinct residues. We have attempted to be explicit and clear that Orb6 phosphorylation of Sts5-S86 was discovered by Chen et al., 2019, while stating that our contribution to understanding this additive mechanism relates to Sts5-S261. We hope that our wording appropriately balances credit for these discoveries, *e.g.* “Recent studies have shown that NDR/LATS kinase Orb6 similarly phosphorylates the Sts5 IDR to regulate its localization to RNP granules (Figure 2A) (Chen et al., 2019; Nuñez et al., 2016). Importantly, Orb6 phosphorylates S86 in the Sts5 IDR (Chen et al., 2019), while we have shown that Pak1 directly phosphorylates S261.”

2. Importantly, the authors refer to "stress granules" in several titles (line 185, line 225, line 244). These statements do not correspond to the data presented and should be corrected. Stress granules (SG granules) are separate membraneless organelles that contain specific factors, differentiating them from P-bodies. The marker Sum2, which is used in the paper, does not identify stress granules, but rather it colocalizes with Dcp1, a component of P-bodies. Therefore, data presented here supports the role of Pak1 in the control of P-bodies, not stress granules.

Thanks for this clarification. Based on this comment, we performed a series of additional colocalization experiments using a stress granule marker Pabp-mRFP. During glucose starvation, Sts5, Pak1, and Sum2 all strongly colocalized with Pabp (new Figure 5 Supp 2). Based on this result, we state in the revised text: “We refer to Pak1 and Sts5 localization to stress granules during glucose deprivation due to colocalization with Pabp, but we note that this result formally supports association with the overlapping stress granule and P body structures.”

3. The 2% glucose control cells used for experiments shown in Figure 4 (Figure 4, Supplement 1) appear very stressed. P-bodies (as visualized by Sum2) do not condense in healthy, exponentially growing cells. While this effect does not invalidate the results of the experiments shown in Figure 4, it would need to be corrected.

These cells were cultured in rich media and displayed no defects in growth rate or other proxies for cell health. Previous studies have shown that P bodies are constitutive structures that can be observed in exponentially growing fission yeast cells (Nilsson and Sunnerhagen, *RNA*, 2011; Wang et al., *Mol Cell Biol*, 2013; Wang et al., *RNA*, 2017) as well as budding yeast cells (Nissan and Parker, *Methods in Enzymology*, 2008). Consistent with these previous studies, we routinely observe P bodies in unstressed cells. These P bodies then become significantly enhanced upon stress such as glucose starvation.

4. Extending the quantification in Figure 7B, to include numbers of Sts5 granules (not only overall fluorescence intensity) would give a more precise assessment of the effects of Pak1 removal on Sts5 granule disassembly.

Thanks for this helpful suggestion. We performed this experiment and added the quantification in the new Figure 8D.

The overall results are interesting and the experiments are solidly performed and analyzed.The paper could be strengthened by genetic epistasis experiments and by following up on the novelty of the Pka1-Pak1 regulatory axis.Comments for the authors:1. Does loss of sts5 suppresses any viability or cell separation defects of pak1 mutants, similarly to what happens in orb6 mutants?

Please see our response to essential revision #5.

2. Following Pak1 kinase activity during glucose deprivation and refeeding could shed light on the dynamics of the Pka1/Pak1 regulatory pathway. Does Pka1 affects Pak1 kinase activity with or without glucose stress and during refeeding?

Please see our response to essential revision #6.

3. Do pak1-CAAX mutants display a delay in resuming polarized cell growth following glucose starvation, as compared with controls? This effect would further support a role for Pak1 in Sts5 dissolution and resumption of translation of specific mRNAs.

Please see our response to essential revision #7.

4. What is the rationale of switching between Tukey's and Dunnett's methods in some of the ANOVA analyses?

We used Dunnett’s one-way ANOVA when comparing every mean measurement to a single control measurement. In contrast, we used a Tukey test to compare each mean to every other mean in the experiment. We have added this information into the Methods section of the revised manuscript.

Reviewer #3:[…] Comments for the authors:1) The usage of ribonucleoprotein (RNP) granulesIn this manuscript, two types of RNP granules are investigated: one is P body and the other is stress granule (SG). P bodies constitutively exist under all conditions, while SGs assemble only under stressed conditions including glucose starvation. It is true that P bodies coalesce with SGs under stresses, but as far as I am concerned, these two types of RNP granules are different. I think it would be better for the authors to use SGs in the text instead of RNP granules when they describe results under glucose deprivation (Figures 4-7). Also, in Figure 7D, RNP should be replaced with SG.

Please see our response to essential revision #1.

2) Binding between Sts5 and 14-3-3Previous work (Nuñez et al., 2016) showed that Orb6-dependent phosphorylation of Sts5 promotes binding between Sts5 and 14-3-3, thereby inhibiting its localization to P bodies. Is this also the case for Pak1-dependent phosphorylation of Sts5? Or in this case, does phosphorylation of serines (S261, S264) within the intrinsically disordered region (IDR) somehow (maybe by disrupting the disordered feature of the IDR?) interfere with the inclusion of Sts5 into P bodies?

Please see our response to essential revision #2.

3) A marker of SGsThis is related to point (1). I think that the authors should have used an authentic marker of SGs, not Sum2 (a marker of P bodies), in experiments under glucose deprivation. At least, co-localization between Sum2 and a marker for SGs under this condition should be shown.

Please see our response to essential revision #3.

4) Dissolution of Sts5 from RNP granules upon glucose re-addition: Figure 7Tethering experiments presented in Figure 7 are of great interest. The difference with regards to the degree of dissolution of Sts5 between wild type cells and those containing CAAX-tethered Pak1 is modest, yet I think it convincing after I scrutinize all the data including those shown in Supplements. I am interested in the physiological consequences, if any, by this tethering. It looks to this referee that cell shape is substantially different between wild type and Pak1-CAAX strains (shown in the bottom row in Figure 7B). I would like to request showing quantification of cell length and width in these cells. I suspect that cells containing Pak1-CAAX display more roundish morphology upon glucose re-addition, which would represent down-regulation of Sts5 function.

Please see our response to essential revision #4.